# Overexpression of Pear (*Pyrus pyrifolia*) *CAD2* in Tomato Affects Lignin Content

**DOI:** 10.3390/molecules24142595

**Published:** 2019-07-17

**Authors:** Mingtong Li, Chenxia Cheng, Xinfu Zhang, Suping Zhou, Lixia Li, Shaolan Yang

**Affiliations:** 1College of Horticulture, Qingdao Agricultural University, 700 Changcheng Road, Chengyang, Qingdao City 266109, China; 2Department of Agricultural and Environmental Sciences, College of Agriculture, Tennessee State University, 3500 John Merritt Blvd, Nashville, TN 37209, USA; 3Dongying Academy of Agricultural Science, Dongying 257091, China

**Keywords:** *PpCAD2*, transgenic tomato, lignin

## Abstract

*PpCAD2* was originally isolated from the ‘Wangkumbae’ pear (*Pyrus pyrifolia* Nakai), and it encodes for cinnamyl alcohol dehydrogenase (CAD), which is a key enzyme in the lignin biosynthesis pathway. In order to verify the function of *PpCAD2*, transgenic tomato (*Solanum lycopersicum*) ‘Micro-Tom’ plants were generated using over-expression constructs via the agrobacterium-mediated transformation method. The results showed that the *PpCAD2* over-expression transgenic tomato plant had a strong growth vigor. Furthermore, these *PpCAD2* over-expression transgenic tomato plants contained a higher lignin content and CAD enzymatic activity in the stem, leaf and fruit pericarp tissues, and formed a greater number of vessel elements in the stem and leaf vein, compared to wild type tomato plants. This study clearly indicated that overexpressing *PpCAD2* increased the lignin deposition of transgenic tomato plants, and thus validated the function of *PpCAD2* in lignin biosynthesis.

## 1. Introduction

During the process of plant growth and development, lignin plays a key role in supporting the plant body, water transport, and resistance against external stress factors [1,2,3,4]. At some fruit postharvest storage stage, the lignin content is an important factor which can affect the fruit texture and quality. In loquat and peach fruit, lignin accumulated during the postharvest storage time, both under room temperature and 0 °C condition [5,6]. In a pear cultivar of *Pyrus pyrifolia*, ‘Whangkeumbae’, hard end is a physiological disorder of the fruit. In the hard end pear fruit, lignin deposits heavily in the pericarp and pulp, leading to the formation of rough textured flesh [7].

Lignin is an amorphous, complex aromatic heteropolymer, which is produced by the phenylpropanoid metabolic pathway [8]. Lignin biosynthesis is a complex process that is divided into three main processes: the biosynthesis of monolignols, transport and polymerization [9]. The monolignols are synthesized from phenylalanine through a series of steps involving phenylalanine ammonia-lyase (PAL), cinnamate 4–hydroxylase (C4H), 4–coumaric acid: CoA ligase (4CL), p–coumarate 3–hydroxylase (C3H), hydroxycinnamoyl: CoA transferase (HCT), caffeoyl–CoA *O*–methyltransferase (CCoAOMT), cinnamoyl–CoA reductase (CCR), ferulate 5–hydroxylase (F5H), Caffeic acid *O*–methyltransferase (COMT), and cinnamyl alcohol dehydrogenase (CAD) [10]. After these steps, monolignols are transported to the apoplast [9]. The monolignols, including p–coumaryl, coniferyl and sinapyl alcohols, are the main building blocks of lignin [2]. Lignin units are polymerized with the monolignols (sinapyl alcohol, S unit; coniferyl alcohol, G unit; and p-coumaryl alcohol, H unit) by peroxidase (POD) and laccase (LAC) [10,11,12]. Gymnosperm lignins consist of G units that only have low levels of H units, and dicotyledonous angiosperm lignins are composed principally of G and S units [2,10].

CAD is a multifunctional enzyme that catalyses the final step in the biosynthesis of monolignols, converting cinnamaldehydes into the corresponding alcohols [2,13]. Related studies have shown that the CAD activity affects not only the content, but also the type of lignin monomer [14]. In our previous work, we have cloned genes of *PpPAL1, PpPAL2*, *Pp4CL1*, *Pp4CL2*, *PpCAD1*, *PpCAD2*, *PpPOD1*, *PpPOD2*, *PpPOD3* and *PpPOD4*, and analyzed their expression patterns in hard end pear fruit [7,15]. The transcript levels of *PpCAD2* were found to have a positive correlation with lignin accumulation in the hard end ‘Whangkeumbae’ pear; the application of calcium chloride alleviated the hard end phenomenon, while simultaneously suppressing the expression of the gene [7]. Studies using transgenic tobacco, poplar, alfalfa, arabidopsis and maize also provided experimental evidences showing that CADs participate in the biosynthesis of the lignin monomer [16,17,18,19,20,21,22,23,24,25]. 

Genetic transformation is a key technology for gene functional verification. For perennial fruit trees, it is hard to carry on the transgenic technology due to the lengthy generation cycles and the difficulty in regenerating in vitro [26]. The agrobacterium-mediated transformation of ‘Micro-Tom’ has been reported in the functional studies of various genes [27,28,29]. The use of tomato to transfer heterogeneous genes from fruit trees, followed by the physiological characterization of the transgenic plants, will greatly reduce the amount of time required to validate the gene function using the fruit tree system. The tomato (*Solanum lycopersicum*) is used as a model plant of the Solanaceae family. A miniature tomato cultivar of *S. lycopersicum*, ‘Micro-Tom’, being an excellent model system, has some characteristics that include a small size, short generation time and a transformable quality, that make it suitable for experimental research [27,30,31]. Recent studies have reported that the cell wall stiffness of tomato fruit skin is mediated by lignin biosynthesis [32,33,34]. It has been reported that inhibiting the lignification of the pericarp may contribute to fleshy fruit during tomato fruit ripening [35]. A MADS-box transcription factor gene, *TOMATO AGAMOUS-LIKE 1 (TAGL1)* controls the lignification of tomato fruit [32]. In *TAGL1*-silenced fruit pericarp, lignin pathway genes, such as *PAL*, *4CL* and *CAD*, are up-regulated, and the lignin content is also increased [32].

In our current study, transgenic tomato ’Micro-Tom’ plants harboring *PpCAD2* constructs were generated via the agrobacterium-mediated transformation method. In transgenic tomato plants, the physiological properties and the expression level of *PpCAD2* were analyzed to clarify the role of *PpCAD2* in lignin synthesis.

## 2. Results and Discussion

### 2.1. Generation of Transgenic Tomato Plants 

To investigate the function of *PpCAD2*, we isolated a 978-bp predicted opening reading frame, encoding a protein of 325 amino acids with a cinnamyl-alcohol dehydrogenase domain (Appendix A). We next overexpressed (OX) *PpCAD2* in tomato. Healthy ‘Micro-Tom’ plants were generated, and these plants were propagated in vitro. The rooted plants were transplanted into pots (Appendix A). The positive overexpressed transgenic tomato lines were confirmed by polymerase chain reaction (PCR) analysis using DNA extracted from mature leaves (Appendix A). Fifteen independent transgenic lines were obtained. Among these, 11 independent lines were selected for RNA extraction. We also evaluated the expression of *PpCAD2* in the *PpCAD2-*-*ox* lines through a qRT-PCR analysis. Compared with the wild type *(WT)* plant, the three *PpCAD2--ox* (#5, #6 and #12) lines exhibited a higher expression of *PpCAD2* (Figure 1B). The three *PpCAD2-*-*ox* (#5, #6 and #12) lines were chosen for the functional analysis (Figure 1A,B). 

### 2.2. The Morphology Indexes of Transgenic Plants Overexpressing PpCAD2 

We compared the morphology of the plants from the WT and the *PpCAD2-ox* lines. Compared with the WT plants, the overexpression of *PpCAD2* resulted in taller plants with a more extensive root system (Figure 1A). The WT plants only had a fewer and shorter roots (Figure 1A). The plant height in the *PpCAD2-ox* lines was significantly higher than for the WT plants (Figure 1C). Like the plant height, the stem diameter in the *PpCAD2-ox* lines was significantly larger than for the WT plants (Figure 1D). However, overexpression of *PpCAD2* did not affect the fruit diameter (Figure 1E). In agreement with our results, in Arabidopsis, the silencing of *CAD C*, and *CAD D* resulted in a severe dwarf phenotype [36]. Moreover, it has been reported that *Medicago truncatula CAD1* mutant *cad1-1* plants exhibit a dwarf phenotype when grown at 30 °C [37]. In rice, *CAD* mutant plants also exhibit a semi-dwarf phenotype [38]. Collectively, these results suggest that *PpCAD2* functions by increasing the plant height and stem diameter but not by affecting the fruit diameter.

### 2.3. Overexprssion of PpCAD2 in Tomato Increased the Lignin Content and CAD Enzymatic Activity in Stem 

The CAD enzyme has been reported to have a key role during the lignin biosynthesis. To determine whether *PpCAD2* regulates the lignin content, we examined the degree of lignification and lignin content in wild-type and *PpCAD2-ox* stems. The Weisner staining results showed that the level of the stem’s lignin staining was higher in the *PpCAD2-ox* lines than in the WT plants (Figure 2A). Overexpression of *PpCAD2* results in more cell layers of xylem elements and a wider xylem tissue (Figure 2B). Observations of autofluorescence also showed that there are more cell layers of xylem elements in the stem of the *PpCAD2-ox* lines than for the wild type plants (Figure 2B). Then, we examined the lignin content and CAD enzymatic activity in the stem of the WT and *PpCAD2-ox* lines. The lignin content of the stem in the transgenic lines overexpressing *PpCAD2* was significantly higher than that in WT (Figure 2C). As expected, the CAD enzymatic activity was significantly higher in the *PpCAD2-ox* lines than for WT (Figure 2D). These results further indicate that the CAD enzymatic activity is positively correlated with the lignin content in the stem. Overexpression of *PpCAD2* increased the level of lignification of the stem tissues, suggesting that *PpCAD2* may play a role in lignin accumulation in tomato stem tissues.

### 2.4. Overexpression of PpCAD2 in Tomato Increased the Lignin Content in Fruit Pericarp and Leaf 

In our previous study, we have proven that CAD is involved in regulating lignin biosynthesis in pear flesh [7]. Previous research had reported that the lignin content could be measured in the pericarp of fruit during tomato fruit ripening [35]. In tomato, the fruit with a higher expression level of the lignin pathway genes, such as *PAL*, *4CL* and *CAD*, had a higher lignin content in the fruit pericarp [32]. To determine the change of the lignin content and CAD activity in the fruit pericarp of transgenic plants, the lignin content and CAD activity were measured. The CAD activity in the fruit pericarp of the *PpCAD2-ox* lines was significantly higher than that of the WT plants (Figure 3A). Similarly, the lignin content of the *PpCAD2-ox* lines was significantly higher than that of the WT plants (Figure 3B). These data suggest that overexpression of *PpCAD2* in tomato increases the CAD activity and the lignin content of the fruit pericarp. 

The degree of lignification in leaf veins was also examined in transgenic plants overexpressing *PpCAD2*. Observations of autofluorescence showed that, similar to the stems, *PpCAD2* transgenic plants had increased layers of xylem elements compared with WT (Figure 4A). In addition, the diameter of the xylem elements was also larger in the *PpCAD2-ox* lines. As with the data in the stem and fruit pericarp, both the lignin content and CAD activity in the leaf veins of the *PpCAD2-ox* plants were significantly higher than those of WT (Figure 4B,C), indicating that overexpression of *PpCAD2* improved the CAD enzyme activity, which in turn enhances the degree of lignification in transgenic tomato leaves. 

*CAD* genes were demonstrated to control lignification in plants [39,40,41]. In agreement with our results, *Brachypodium distachyon CAD* gene *BdCAD1* mutants displayed a reduced CAD activity and lower lignin content [39], suggesting that the CAD activity is positively correlated with the lignin content. In *Brassica chinensis*, the induced expression of *BcCAD1-1* and *BcCAD2* could increase the lignification of stems [40]. *AaCAD* has also been reported to positively enhance lignin formation in *Artemisia annua* [41]. In summary, these results support that *PpCAD2* positively regulates lignin biosynthesis by increasing the CAD activity in the stems, fruit pericarp and leaves of transgenic tomato plants. In pear, the lignin content and the expression level of *PpCAD2* were positively correlated with the occurrence of fruit with hard end [7]. From the above results, we infer that the down-regulation of *PpCAD2* expression could appropriately inhibit the formation of fruit hard end by decreasing the lignin content. Further investigations to clarify the role of *PpCAD2* in the occurrence of fruit with hard end and to determine the upstream regulatory factor of *PpCAD2* will provide a better mechanistic understanding of the formation of fruit with hard end. 

## 3. Materials and Methods

### 3.1. Plant Material

Tomato (*S. lycopersicum*) ‘Micro-Tom’ seeds were disinfected by submerging in 75% ethanol for 2 min followed by 10% sodium hypochlorite for 10 min. After three rinses with sterile water and being blotted dry of the excess water on the seed surface, the seeds were inoculated on a Murashige and Skoog (MS) medium. The cultures were kept in the dark and transferred to light conditions when cotyledons emerged [42,43]. 

### 3.2. Vector Construction and Tomato Transformation

The primers with restriction enzymes sites were designed using DNAStar software (Lasergene). *PpCAD2* over-expression primers, forward: 5’CGTCTAGAAGATGAGCAGCGGAGCAG (*Xba* I) 3’, reverse: 5’ CCGGATCCAAGGGAAGCCGGAGTTTA (*BamH* I) 3’. The full-length cDNA of *PpCAD2* was ligated into a cloning vector pMD19-T. To prepare the *PpCAD2* over-expression (*35S::PpCAD2*) constructs, plasmids were isolated and digested with the restriction enzymes *BamH* I and *Xba* I. The insert fragment was isolated and then cloned onto a vector, pBI121, under a 35S promoter. These constructs were transformed into *Agrobacterium tumefaciens* EHA105 through the freeze-thaw method separately [44]. 

For the overexpression, cotyledons of ‘Micro-Tom’ were transformed by an Agrobacterium-mediated transformation. The cotyledons of ‘Micro-Tom’ were sectioned to 3 mm^2^, and one hundred cotyledon explants were dipped into a bacterial suspension (OD_600_ = 0.3–0.4) of *A. tumefaciens* EHA105 harboring the transformation constructs. After 5 min, the explants were blotted dry with autoclaved filter paper. The explants were placed back onto the co-cultivation medium and incubated in the dark for 1.5 d under 28 °C. The explants were transferred to a callus induction/selection medium containing MS salts, 3% sucrose, 1.5 mg·L^−1^ zeatin, 50 mg·L^−1^ kanamycin and 500 mg·L^−1^ Cef, and 0.7% agar at pH 5.8 [41]. The explants were transferred onto a fresh medium every 2 weeks. When adventitious buds developing from the callus grew to about 3 cm tall, they were dissected and transferred to a new conical flask containing a shoot elongation medium containing MS salts, 3% sucrose, 1.0 mg·L^−1^ 6–Benzylaminopurine (6–BA), 0.1 mg·L^−1^ Indole-3–Butytric acid (IBA) and 0.7% agar. The shoots forming a few leaves were transferred to a rooting medium containing MS salts, 3% sucrose, 0.2 mg·L^−1^ IBA and 0.7% agar. The rooted plants were transplanted to pots and grown in a growth chamber that was programmed at a constant temperature of 21 °C and a 16-h-light/8-h-dark cycle. Wild type (WT) plants were propagated concurrent to the transgenic plants. 

### 3.3. PCR Analysis and Gene Expression Analysis

Genomic DNA was extracted from ‘Micro-Tom’ leaves using a plant total DNA extract kit (TianGen, Shanghai, China). The PCR amplification used primer pairs of *PpCAD2* forward: 5’-CGTCTAGAAGTAGAGCAGCGGAGCAG-3’, and *PpCAD2* reverse: 5’-CCGGATCCAAGGGAAGCCGGAGTTTA-3’. 

Total RNA was extracted from the ‘Micro-Tom’ leaf tissue using EASYspin Plant RNA Kit (Yuanpinghao, China) according to the manufacturer’s instructions. Genomic DNA was removed by DNase. cDNA was synthesized with reverse transcription using the Prime Script™ RT reagent Kit (Takara, Dalian, China) according to the manufacturer’s instruction and was used as the template for the qPCR analysis. qPCR was performed on a Light Cycler® 480 instrument (Roche, Basel, Switzerland). The procedure included annealing at 94 °C for 5 min, followed by 40 cycles of 94 °C for 15 s, 60 °C for 1 min, and 72 °C for 30 s. A tomato actin gene was used for normalization. The primers used for the qPCR analysis, listed in Table 1, were designed using Primer 3 (http://bioinfo.ut.ee/primer3-0.4.0/). The relative gene expression level was calculated using the 2^-ΔΔCt^ method [45]. Three biological replicates were performed for each sample.

### 3.4. Lignin Content Determination

The lignin content of the stem, leaf and fruit pericarp in the three-month-old ‘Micro-Tom’ tomato was measured according to the method described by Dyckmans [46]. Frozen tissue powder (200 mg) was suspended in 10 mL of washing buffer (100 mM K_2_HPO_4_/KH_2_PO_4_, 0.5% Triton X-100, 0.5% PVP, pH 7.8) for 30 min. After being centrifuged, the pellet was washed twice (30 min) in 100% MeOH. Then, the resulting pellet was dried in 80 °C overnight. The dried pellet was added with 1 mL 2M HCl and 0.1 mL thioglycolic acid, then put into boiling water for 4 hours. The obtained end product was dissolved in 1 mL 1M NaOH. The diluted samples were assayed for absorbance at 280 nm, and NaOH was used as a blank. All measurements were performed in biological triplicates.

### 3.5. Determination of Biomass Parameters

The plant height of the three-month-old ‘Micro-Tom’ tomato was measured with a ruler. Additionally, the stem diameter and mature fruit diameter were measured with a vernier caliper. Three biological replications were determined.

### 3.6. CAD Enzyme Activity

The stem and fruit CAD enzymatic activity was measured according to the method described by Cai [47]. The samples were grinded with 10 mL Tris:HCl buffer (200 mM, pH 7.5). The reaction mixture contained 50 μL of extract, 1 mL of 100 mM Tris:HCl buffer (pH 8.8), 1 mL of 20 mM coniferyl alcohol and 1 mL of 5mM NADP+. The mixture was put at 37 °C during 2 min for the reaction, after which 0.5 mL 1 mol·L^−1^ HCl was added to terminate the reaction. The samples were assayed for absorbance at 400 nm. The data were expressed on a protein basis, and the analysis was biologically repeated three times.

### 3.7. Weisner Staining and Microscopy

The transverse section of the leaf veins and stem were used by Weisner staining and microscopy. Weisner reagent (phloroglucinol/HCl) was used to stain the plant tissue for 5 min before it was visualized for lignification under a microscope [48]. The lignified structures appeared pink or fuchsia in the bright field images. Auto-fluorescence within the leaf and stem sections was also observed with the aid of an EVOS smart fluorescence microscope (Thermo Fisher, Waltham, MA, America).

## 4. Conclusions

In this study, overexpression of *PpCAD2* in transgenic tomato plants increased the plant height and stem diameter. Furthermore, overexpression of *PpCAD2* increased the lignin content in the stems, leaves and fruit pericarp tissues, partially by increasing the CAD enzyme activity, which was an important enzyme in the biosynthesis of monolignols. Overexpression of *PpCAD2* also increased the size of the vessel element in the xylem tissues. Our data suggest that the *PpCAD2* functions by positively regulating the degree of lignification. 

## Figures and Tables

**Figure 1 molecules-24-02595-f001:**
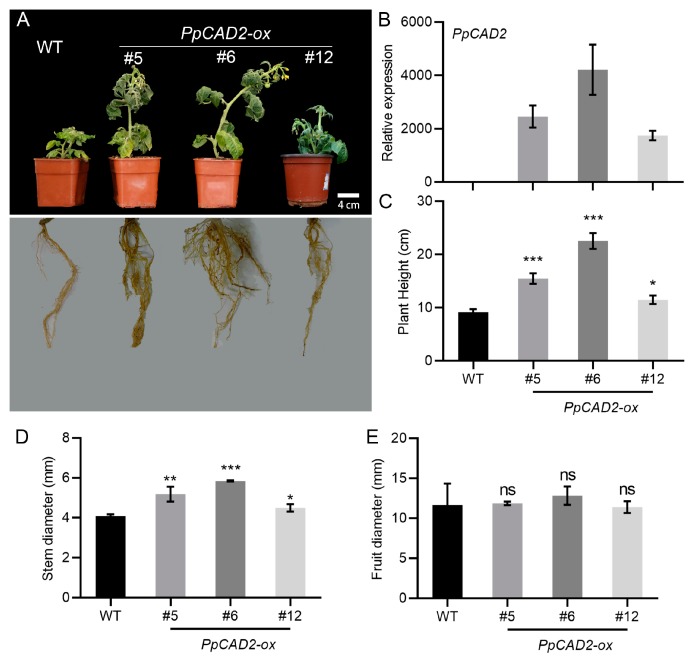
Phenotype analysis of non-transgenic (WT) and *PpCAD2*-overexpressing (*PpCAD2-ox*) transgenic tomato lines. (**A**) Photographs of the morphology in the WT and transgenic lines; (**B**) The relative expression level of *PpCAD2* in the WT and transgenic plants; (**C**) The plant height of the WT and transgenic plants; (**D**) The stem diameter the WT and transgenic plants; (**E**) The fruit diameter of the WT and transgenic plants. In all cases, the data represent the mean ± SD (standard deviation) from three biological replicates. Significant differences between the wild-type and transgenic plants are indicated (**P* < 0.05, ***P* < 0.01, ****P* < 0.001, Student’s *t-test*).

**Figure 2 molecules-24-02595-f002:**
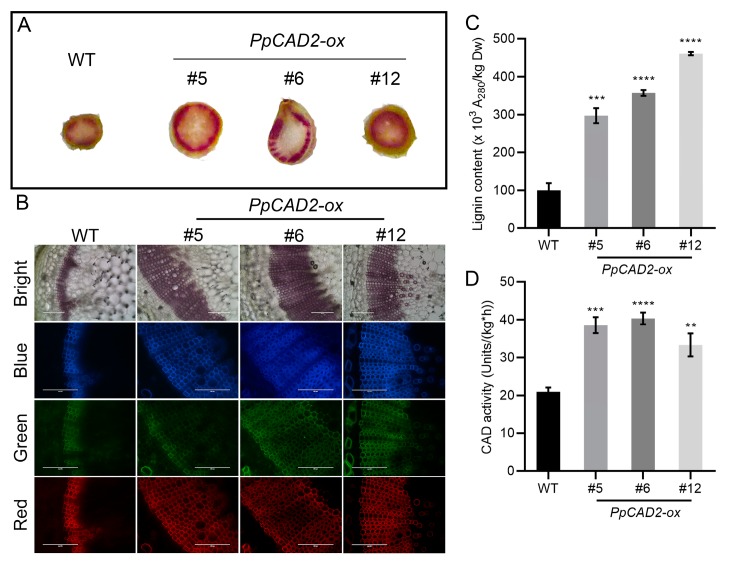
*PpCAD2* increases the lignin content and the cinnamyl alcohol dehydrogenase (CAD) enzyme activity in the stem of transgenic tomato plants. (**A**) Transverse sections of the stem were stained with phloroglucinol–HCl for the detection of lignin; (**B**) Autofluorescence of the stem’s transverse slice. Bright: bright field images, Blue: blue autofluorescence, Green: green autofluorescence, Red: red autofluorescence. Scale bars = 200 μm; (**C**) The lignin content of the stem in the WT and transgenic plants; (**D**) The CAD enzyme activity in the stem of WT and transgenic plants. In (**C**) and (**D**), the data represent the mean ± SD (standard deviation) from three biological replicates. Significant differences between the wild-type and transgenic plants are indicated (***P* < 0.01, ****P* < 0.001, *****P* < 0.0001, Student’s *t-test*).

**Figure 3 molecules-24-02595-f003:**
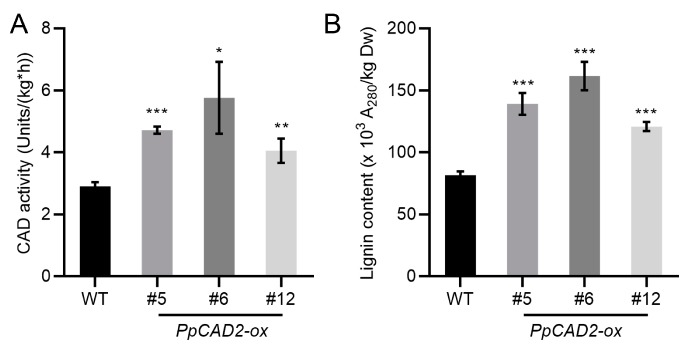
*PpCAD2* increases the lignin content and CAD enzyme activity in the fruit pericarp of transgenic tomato plants. (**A**) The lignin content of the fruit pericarp in the WT and transgenic plants; (**B**) The CAD enzyme activity in the fruit pericarp of the WT and transgenic plants. In (**A**) and (**B**), the data represent the mean ± SD (standard deviation) from three biological replicates. Significant differences between the wild-type and transgenic plants are indicated (**P* < 0.05, ***P* < 0.01, ****P* < 0.001, Student’s *t-test*).

**Figure 4 molecules-24-02595-f004:**
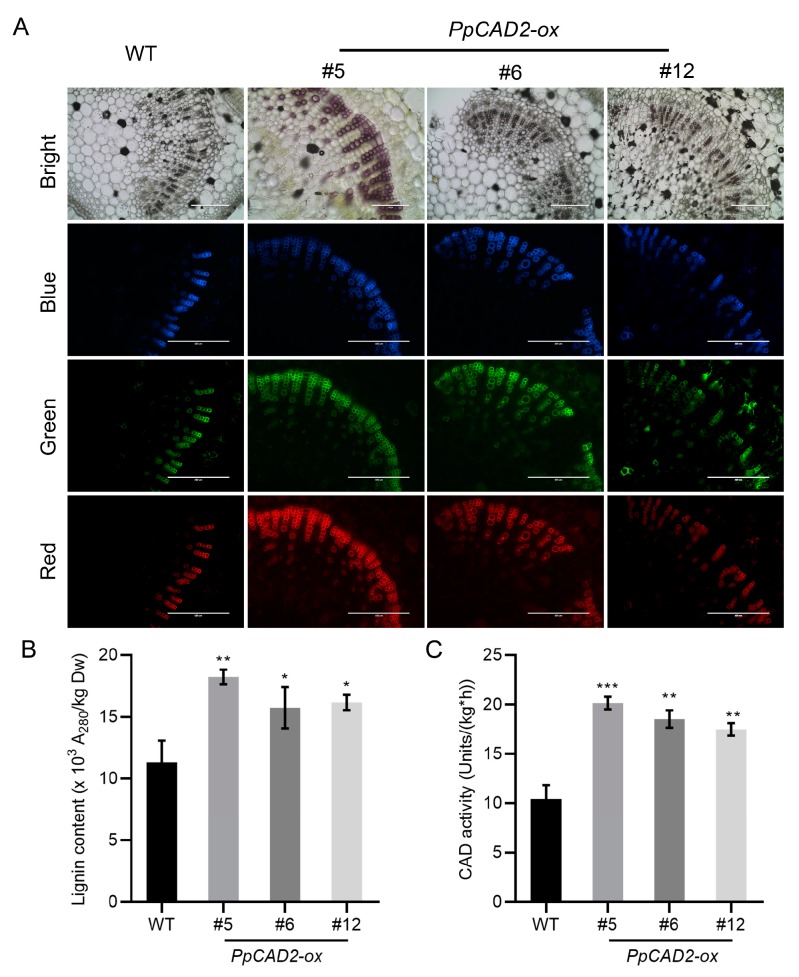
*PpCAD2* increases the lignin content and CAD enzyme activity in the leaves of transgenic tomato plants. (**A**) Autofluorescence of transverse slice in the leaf veins, Bright: bright field images, Blue: blue autofluorescence, Green: green autofluorescence, Red: red autofluorescence. Scale bars = 200 μm; (**B**) The lignin content of leaves in the WT and transgenic plants; (**C**) The CAD enzyme activity in the leaves of WT and transgenic plants. In (**B**) and (**C**), the data represent the mean ± SD (standard deviation) from three biological replicates. Significant differences between the wild-type and transgenic plants are indicated (**P* < 0.05, ***P* < 0.01, ****P* < 0.001, Student’s *t-test*).

**Table 1 molecules-24-02595-t001:** Primers used for the q-PCR analysis.

Gene Name	Gene ID	Primer Name	Primer Sequence (5’ to 3’)
*PpCAD2*	KJ577637	PpCAD2-F	TTTGGTTGAGAGAGTTGCCCAC
		PpCAD2-R	ATTCGACACCCAAGCTCTTCG
*SlActin*	LOC101264618	SlActin-F	CAGATGTGGATAACGAAGGCC
		SlActin-R	TCACAGTAGAAAGACCTGAACAA

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
