# Peer review of "Overexpression of Pear (Pyrus pyrifolia) CAD2 in Tomato Affects Lignin Content"

_molecules, 2019, doi:10.3390/molecules24142595_

Round 1

Reviewer 1 Report

This revised version of the manuscript shows an overall improvement. The points raised by myself and the other reviewers are for the most part addressed properly. The new manuscript is easier to follow by the readership and does not make claims that not supported by the data, as before. I especially appreciated the change in the title and the decision to remove the results of the silencing approach.

Author Response

Comments and Suggestions for Authors

This revised version of the manuscript shows an overall improvement. The points raised by myself and the other reviewers are for the most part addressed properly. The new manuscript is easier to follow by the readership and does not make claims that not supported by the data, as before. I especially appreciated the change in the title and the decision to remove the results of the silencing approach.

Response: We thank the reviewer for the detailed and thorough reviews.

Reviewer 2 Report

Manuscript ID: molecules-540924

Overexpression of pear CAD2 in tomato affects lignin content by Li et al.

In the presented study authors analyzed the function of key enzyme in lignin biosynthesis pathway, CAD2, encoding for cinnamyl alcohol dehydrogenase (CAD). This gene was originally isolated from pear (Pyrus pyrifolia Nakai) and overexpressed in tomato ‘Micro-Tom’ plants via agrobacterium-mediated transformation method. As authors claim, PpCAD2 over-expressing transgenic tomato plants had strong growth vigor, accumulated higher lignin content and exhibited higher CAD enzymatic activity in stem, leaf and fruit pericarp tissues, compared to wild type tomato plants.

The scientific approach of presented study employing integration, overexpression and  functional characterization of selected gene in transgenic tomato is applicable. Manuscript is well written with adequate literature referred to, results are in general presented in appropriate manner.

However, I found some lacks in presenting results related to genetic transformation procedure and confirmation of transgene integration, as well as in gene expression analysis. Also, authors are suggested to provide information about statistical methods used for data analyses. In addition, the general aim of the submitted manuscript cannot be only validation of already known physiological role of CAD. Authors should propose some more farsighted prospects of obtained results that will be attractive for biotechnology implementation in for example fruit quality assessments.   

Taking all together, I regret to tell that the manuscript in its present form and with this unconvincing PCR and gene expression analysis results is not ready for publication and revision in results interpretation is required.

Specific comments on the manuscript:

Line 76, Line 188: Please specify how many explants were used for transformation. Related to this, overall transformation efficiency is important parameter for genetic transformation experiments, so please add it to the results section.

Line 79: I believe plants were propagated “in vitro” instead “in vivo”. 

Line 81, Line 207: Authors used pPCR 2-ΔΔCt method to measure PpCAD2 expression levels in transgenic tomato lines. Using this method expression level in one sample is calculated relative to sample designated as calibrator (usually control treatment). As authors wrote in Material and methods and shown on Fig. 1, WT plants were set as calibrator and expression in transgenic lines were presented relative to expression of analyzed gene in WT. How it is possible since WT plants did not have integrated PpCAD2 gene (PCR gel confirms that) and therefore expression of this gene in WT plants could not be detected. qPCR absolute quantification is thus valid method for determination  the transcripts copy number obtained from standards with known numbers of gene transcript copies. Please, provide explanation or replace provided results presented on Fig. 1B with absolute quantification of transgene.

Line 191: Replace word “to fresh plate” with “onto fresh medium”

Line 193: What was composition of shoot elongation medium?

Line 196: I believe the number of hours of light should be 16 instead of 14.

Line 197: Add abbreviation (WT) after “wild type”.

Line 199, Figure S3: Please specify how many transgenic lines were subjected to PCR transgene integration confirmation. All lines survived selection procedure? Could authors provide uncroped picture with positive control, blank and DNA marker shown on it?   

Line 203: Add quotation marks to Micro-Tom.

Line 208: Please add the elongation phase in description of PCR cycling.

Line 209: According the gene ID, used actin is from pear, not from tomato. Also, in previous paper of the same group of authors (J. Plant Grow. Reg. 2015) the same gene with the same primers is referred to pear. Please check.

Line 212: I think that biological replicates were performed for each “sample” not a “gene”.  

Line 214, 219, 222, 226: Please provide a bit more information about all used methods. For example, it is not clear in which plant parts lignin content was determinate, how old were plants subjected to morphological measurements, what tissue was used for microscopic lignin visualization.

Author Response

This manuscript is a resubmission of an earlier submission. The following is a list of the peer review reports and author responses from that submission.

Round 1

Reviewer 1 Report

The manuscript of Li et al presents the characterization of the pear CAD2 gene in tomato, a heterologous system. Because pear is a relatively important crop (especially in China) and because lignification of stone cells is a major factor affecting pear fruit quality, it is quite important to study lignin metabolism in this species. However, the data presented in this manuscript are not solid. First, I don’t understand exactly why the authors carried out a strategy of gene silencing in a heterologous system; I mean, of course that overexpressing the gene is a valuable strategy, but silencing? What is the output of that strategy? Most likely the orthologous gene will be silenced, but then, how much can you say about the function of the original gene? The same information can be gained simply by identifying an orthologous gene in a plant species for which functional studies have been carried out; Noteworthy, the antisense plants do not show a lignin phenotype. Second, the analysis of the vasculature is not quantitative, so it is not supported by the data. Third, lignin content was not measured in stems (which is a major lignification site) but it was measured in tomato fruits (that barely deposits lignin). Fourth, the authors analyzed only the expression of one tomato CAD gene, but their strategies might affect several CAD genes in tomato. So it is currently impossible to know. 

Title: by using PpCAD2 it is impossible to know from which plant species this gene is coming from; I suggest changing the title to “Overexpression of pear CAD2 in tomato affects lignin content” 

Abstract:

-  Line 18: why the abbreviation of the tomato gene is LeCAD1 and not SlCAD1? Please correct

- Lines 18-20: this sentence is redundant with the previous one regarding the information on lignin content; please rephrase

- Line 23: I don’t think “improved” is the best expression here; I would rather say “increased lignin deposition”

Introduction

- Lines 29-30: is lignin important for the quality of all fruits? Most fruits do not deposit significant amounts of lignin, so isn’t this sentence for specific for some types of fruits, such as pears that have stone cells?

- Line 30: change “index” to “factor”

- Line 32: I really don’t understand what “In the hard end pear fruit” means; can the authors explain that?

- Line 34: change “phenylpropane” to “phenylpropanoid”

- Lines 34-38: this paragraph is poorly written, with very short and poorly informative sentences and with serious mistakes regarding general information on lignin metabolism; 1) those are not lignin monomers; lignin monomers are the hydroxycinammyl alcohols p-coumaryl, coniferyl and sinapyl alcohols, also known as monolignols; these are lignin units, which result from the incorporation of the monolignols (and the correct is p-hydroxyphenyl); 2) G and S are the major lignin units found in the lignin of Angiosperms; 3) Lignin biosynthetic pathway involves at least 11 steps and not only the 3 genes listed here; peroxidases are NOT part of the lignin biosynthetic pathway; 4) finally, the authors do not provide enough information on lignin metabolism for the reader; please, provide more information

- Lines 50-53: this paragraph is also poorly written; please rephrase it (and make it clear why to use tomato in your study with pear genes)

- Lines 54-58: it is not clear why the authors chose specifically PpCAD2 for their functional studies; in a previous paragraph, they state that they have cloned and analyzed the expression of several lignin biosynthetic genes; in addition, they claimed that both PpCAD1 and PpCAD2 show an interesting expression pattern; why PpCAD2 then?

Results & Discussion

- Lines 61-67: this should be included in the introduction (please note that I made a comment asking about the lack of information on why to use tomato on your study)

- Lines 68-72: this is material and methods and figure 1 and 2 should not be included in the main manuscript (it should be included as supplementary)

- Lines 78-80: as the authors performed silencing in a heterologous system, how come they analyze the expression of PpCAD2 in tomato in the silencing strategy? Isn’t this the expression analysis of the corresponding tomato gene? Please clarify

- Lines 85-86: this sentence is redundant with the following sentence (lines 88-90); please merge the same information onto one sentence

- Table 1: for me it seems very weird that the overexpressing plants are taller than the WT; as there is no information on how the biomass parameters were assessed and how the data was analyzed, but for me there is no difference in plant height between OE lines and WT; can the authors clarify that?

- Figure 4a: I don’t think the information on the expression level of PpCAD2 should be included in the main manuscript; this should be only the parameter to select which transgenic line to select for further characterization

- Line 117-118: I don’t understand this sentence; I mean, the authors are using a heterologous system, so how come can they evaluate the expression of the heterologous gene in tomato?!?!? I doesn’t make sense

- Lines 120-122: why did the authors specifically chose SlCAD1 as an endogenous tomato gene? Normally the CAD family includes several paralogs so it would be important to evaluate all tomato paralogs, especially for the antisense strategy; this is a very important issue

- Lines 132-140: why did the authors perform lignin quantification experiments in leaves and fruits but not in stems? Stem tissue is considered the major lignification site within a plant body; additionally, where exactly lignin is deposited in tomato fruit? What is the premise in analyzing lignin content in a tissue that barely deposits lignin?

- Figure 6: why did the authors “pool” the different transgenic lines in these analyses? The authors should present the data for each individual line

- Lines 168-174: the authors have no solid data to state that there are more xylem vessels produced in their lines; no quantitative analysis was perform and I don’t think they can make such conclusions

Material and Methods

- there is no information on how the tomato plants were phenotyped for biomass parameters

Reviewer 2 Report

Authors have identified and characterized Cinnamy Alcohol Dehydrogenase2 (CAD) from Pyrus pyrofolia (Pp) by expressing it in tomato. 

-Authors have not given any sequence or phylogenetic analysis for CAD isoforms of Pp. It is difficult to judge why particularly CAD2 isoform was chosen and what is the function of CAD2. 

 -I don’t see it is necessary to show Figure 2 in this manuscript. If it is the author has to explain why 2ndand 6thwell did not show any band. Was it not expressed at all? 

-Authors have observed height, stem, fruit phenotype but this needs to backed up by concrete data. It will easier to conclude about if author mentions in how many transgenic lines they observed this effect on phenotype? Also, statistics are not mentioned in any of the figures which makes it difficult to conclude about the phenotype. Also, author should discuss why they observed height phenotype? Was it because of an increase in cell size or cell elongation? Was it observed in any of the other CAD mutants?

-Figure 4: Author should mention what are the error bars and how many biological replicates were used for the experiment. Also, what is the expression of PpCAD1 

-Figure 5 and 6: Please explain the unit for enzyme activity and lignin content. It is easier for the reader if author represents lignin content as mg/g or %. 

 -Figure 7 and 8: Did author tried Maule staining? That staining would whether these is any effect on lignin composition. 

Overall the discussion about the phenotype is lacking in this paper. Lignin pathway is well studied and comparison with other results would educate the reader about the novelty of this paper. 

Reviewer 3 Report

In this manuscript, Li and co-authors describe their study on characterization of  the PpCAD2 enzyme  in the context of  lignin synthesis. Overall, the manuscript lacks clarity of the aims and structure.

There is a number of issues that require extensive editing:

-       first, the  reason why Micro Tom tomato was chosen as a model plant. Tomato produce berries while pear fruits are syncarpous fleshy fruits.

-        It is unclear why a silencing approach was attempted by means of a silencing construct against a sequence from a different species. Is the sequence sufficiently similar? An alignment should probably be shown.

-        Why was the LeCAD1 antisense not chosen?

-        All the information concerning the statistical analysis carried out on all datasets are  missing.

-        The LeCAD1 relative expression levels in S6-PpCAD2 transgenic plant is significatively higher than Wt plants. An explanation for this aobservation should be provided.

Minor points:

Lines 52-53: Specify the type of fruit features.  

Lines 69-70: PCR analysis using DNA extracted from mature leaves confirmed the positive over-expression”     By PCR analysis on genomic DNA one can only confirm the presence of the T-DNA inside the genome, but not information over expression levels.

Line 71: Which generation of transgenics were these?

Line 230: Are these biological replicates?

Line 236: “All measurements were performed in triplicates”. Concerning lignin quantification in fruits, clarify whether these were technical or biological replicates.  Also, please describe better how the sampling was conducted.

Table 1: Please specify the number of replicates

Figure 5 and 6: Uniformity is required for figure legends in axis.

Finally, a thorough revision of the English language is recommended.